# Sex differences in the prevalence of metabolic syndrome and associated factors in the general population of Mongolia: A nationwide study

Lkham-Erdene Byambadoo[1,2], Narantuya Davaakhuu[3], Unursaikhan Surenjav[3], Bolormaa Norov[3], Enkhjargal Tserennadmid[3], Souphalak Inthaphatha[1,4], Kimihiro Nishino[1], Eiko Yamamoto[1] *

1 Department of Healthcare Administration, Nagoya University Graduate School of Medicine, Nagoya, Aichi, Japan, 2 Department of Public Health Policy, Ministry of Health, Ulaanbaatar, Mongolia, 3 National Center for Public Health, Ulaanbaatar, Mongolia, 4 Asian Satellite Campuses Institution, Nagoya University, Nagoya, Aichi, Japan

* yamaeiko@med.nagoya-u.ac.jp

**Data Availability Statement:** All data files are available from the World Health Organization NCD Microdata Repository after user registration and

## Abstract

The prevalence and factors associated with metabolic syndrome (MetS) between men and women in Mongolia were compared using secondary data from the STEPwise approach to non-communicable disease risk factor surveillance conducted in 2019. In total, 5,695 participants (2,577 males and 3,118 females) aged 18–69 years old were enrolled in the study. The prevalence of MetS was 37.4% in total and the prevalence was significantly higher in women (39.2%) than men (35.2%, P = 0.002). The most prevalent MetS components were abdominal obesity in women (74.2%) and high blood glucose levels in men (58.7%). All of the 2,128 participants who were categorized into MetS met the criteria of abdominal obesity. Factors associated with MetS included being 30–69 years old compared to 18–29 years old, low and moderate physical activity levels compared to high levels, history of hypertension and diabetes mellitus, and a high body mass index (overweight and obesity) compared to a normal body mass index in both sexes. Additional factors included Buryat ethnicity compared to Khalkh ethnicity, living in Ulaanbaatar, no education compared to primary education, higher monthly income, and currently drinking in men; and smoking and sufficient fruit and vegetable intake in women. Significant differences were observed between men and women in terms of the prevalence of MetS, components of MetS, and factors associated with MetS. To reduce MetS prevalence in Mongolia, sex-specific programs should be designed to promote health behavior, such as reducing alcohol consumption in men and smoking in women.

## Introduction

Metabolic syndrome (MetS) is characterized by an interrelated combination of cardio-metabolic risk factors that include abdominal obesity, hypertension (HT), hyperglycemia, decreased

getting approval for dataset access (https://extranet.who.int/ncdsmicrodata/index.php/catalog/836).

**Funding:** The author(s) received no specific funding for this work.

**Competing interests:** The authors have declared that no competing interests exist.

high-density lipoprotein cholesterol (HDL-C) levels, and elevated triglyceride (TG) levels [1]. MetS is a substantial contributor to premature mortality and disease burden worldwide and a major public health problem in both developing and developed countries [2–4]. Having multiple MetS components increases the risk of cardiovascular disease (CVD), type 2 diabetes mellitus (DM), various cancers, and mortality related to these conditions [5,6]. Approximately 20–25% of adults are diagnosed with MetS globally [1], but the prevalence of MetS in each country varies depending on the lifestyle, ethnicity, and proportion of the aging population. Socio-demographic factors (female, ethnicity, older age, low socioeconomic status, and living in urban areas) and health behavior and lifestyle factors (physical inactivity, smoking, drinking alcohol, unhealthy eating, and overweight and obesity) have been identified as MetS-related external environmental factors [7–10]. A systematic review on MetS reported that there were significant differences in the prevalence of MetS and risk factors between men and women [11].

Mongolia is a lower middle-income country in East Asia with a population of 3.4 million, two-thirds of whom live in urban areas. Mongolian people have traditional nomadic lifestyles, with a meat-oriented diet, common drinking culture, and an environment that discourages physical activity (PA) owing to extreme weather conditions [12]. Life expectancy in Mongolia increased greatly after 1995, especially for women. The difference in life expectancy between men and women also increased, reaching nine years in 2019 [13]. Of all deaths in 2016, 80% were caused by non-communicable diseases (NCDs), including CVD and DM, and CVD has been the major cause of death in the past three decades [14,15]. Previous studies suggested that the prevalence of MetS in Mongolia has been increasing but the characteristics of the study subjects and the definition of MetS were different among the studies: 12.0% among people aged 30–60 years in 2005, 32.8% in the general population aged 40 years or more without ischemic heart disease in 2015, and 58.0% in adults $\geq$ 20 years old in rural areas in 2019 [12,16,17]. Pengpid et al reported that the prevalence of MetS in the Mongolian adult population increased from 33.3% in 2009 to 37.0% in 2019 using the same MetS criteria, but the criteria used was not the international standard [18]. Factors associated with MetS were reported to include being female, older age, living in the Khangai region, being a widow, moderate or high level of alcohol consumption, moderate or high intensity of regular exercise, and depression [12,17]. In the healthcare setting, it is important to understand the sex patterns in MetS and the associated factors for identifying vulnerable populations and establishing prevention programs [19]. Therefore, the aim of this study was to compare the prevalence and factors associated with MetS between men and women in Mongolia using representative data of the entire population and the international standard criteria of MetS.

## Material and methods

### STEPwise approach to NCD risk factor surveillance (STEPS)

This is a cross-sectional study using secondary data from the STEPS survey that was conducted in Mongolia in 2019. STEPS was a nationwide survey that included 6,654 participants from all 21 provinces and nine districts of Mongolia and was conducted by the National Center for Public Health and the Ministry of Health with technical support from the World Health Organization (WHO) [20].

In brief, this survey was designed to cover all geographic areas of Mongolia, and a multi-stage and stratified sampling procedure was conducted to ensure that the survey population would provide a reliable estimate. To obtain nationally representative estimates, sampling weights were used. The entire country was stratified into urban and rural areas, and the sample was drawn proportionally to the target population in each area. Ulaanbaatar and the capitals of

the 21 provinces represented the urban areas, whereas the remaining soums (a subdivision of a province) represented the rural areas. The survey covered 377 primary sampling units, including kheseg (a subdivision of a district) from urban areas and bag (a subdivision of a soum) from rural areas. Subsequently, 18 households per unit were selected using systematic random sampling. One participant (15–69 years old) was randomly selected from each household for data collection. To ensure data accuracy, non-response cases were considered for bias prevention using weight calculations. The inclusion criteria for the STEPS were individuals aged 15–69 years who agreed to participate in the survey. In total, 6,654 participants participated in the survey between June and September 2019. Trained interviewers collected the data. A modified WHO STEPS questionnaire was used to collect socio-demographic and behavioral data. Standardized and specific equipment and techniques were used for anthropometric measurements, and peripheral blood, serum, and spot urine were collected for laboratory analyses.

## Study variables and measurements

This study included socio-demographic factors, health behavior and lifestyle factors, and MetS measurement data of 5,695 adult participants of the STEPS who were 18–69 years old and whose data had no missing values for variables related to the MetS criteria.

Socio-demographic factors included sex (male or female), age (18–29 years, 30–44 years, and 45–69 years), ethnicity (Khalkh, Kazakh, Durvud, Buryat, or other), place of residence (urban or rural), region (western, eastern, Khangai, central, or Ulaanbaatar), education (no education, primary, secondary, or college and higher), marital status (never married, currently married, or divorced/widowed/separated), employment (full-time, part-time, or unemployed), and monthly income ($< 100,000$ Mongolian tugrik [MNT], $100,000 - < 300,000$ MNT, $300,000 - < 500,000$ MNT, $500,000 - < 1,000,000$ MNT, or $\geq 1,000,000$ MNT). Health behavior and lifestyle factors were current smoking status (no or yes), current alcohol consumption status (no or yes), insufficient intake of fruits and vegetables (no or yes), level of PA (high, moderate, or low), sedentary lifestyle (no or yes), history of HT, history of DM, history of hypercholesterolemia (HCE), history of CVD (no or yes), and body mass index (BMI) (normal, underweight, overweight, or obesity). The definitions of health behavior and lifestyle variables are summarized in Table 1.

MetS measurement data included waist circumference, TG, HDL-C, blood pressure, and fasting blood glucose (FBG). The total number of STEPS participants was 6,654 but 959 participants were excluded due to missing values on MetS measurement data. Finally, the data of 5,695 participants were analyzed in this study. There were missing values about ethnicity (n = 26), education (n = 1), marital status (n = 7), employment (n = 120), fruit and vegetable intake (n = 240), physical activity (n = 88), and body mass index (n = 1). The missing values were treated by multiple imputation method [21].

## Definition of MetS

MetS was defined according to the International Diabetes Federation [1]. The definitions of MetS are listed in Table 1.

## Statistical analysis

A chi-square test was used for comparing characteristics between males and females. A logistic regression analysis was performed to identify factors associated with MetS. In multivariable logistic regression analysis on MetS, three models were performed. In Model 1 (the forced-entry method), all the variables other than abdominal obesity were included. In Models 2 and 3 (the forward and backward stepwise selection method), variables were added when the P-

**Table 1. Definitions of variables.**

| Variable | Definition |
|---|---|
| Metabolic syndrome (International Diabetes Federation criteria) | WC $\geq$ 90 cm among men or $\geq$ 80 cm among women (according to the guide for the Asian population); plus, any two of the four criteria below:<br>1. BP: $\geq$ 130 mmHg systolic or $\geq$ 85 mmHg diastolic or currently taking antihypertensive medicine<br>2. FBG: $\geq$ 5.6 mmol/L or currently taking medication<br>3. Serum TG: $\geq$ 1.7 mmol/L or currently taking medication<br>4. HDL-C< 1.03 mmol/L in men or < 1.29 mmol/L in women or currently taking medication |
| Currently smoking | Smoked in the previous 30 days. |
| Currently drinking | Consumed alcohol in the previous 30 days. |
| Insufficient fruit and vegetable intake | An intake of less than five servings of fruits and vegetables per day. |
| Level of PA | The level of PA was determined using the GPAQ. Responses were converted to MET-min/week according to the GPAQ scoring protocol. The protocol also provides details for data processing, cleaning, and truncation. The MET-min/week spent on each domain was subsequently computed to obtain the overall PA level. High PA level was defined as vigorous-intensity activity on at least three days with at least 1,500 MET-min/week or more on any combinations of walking, moderate or vigorous intensity activities of at least 3,000 MET-min/week. Moderate PA level was defined as three or more days of vigorous-intensity activity of at least 20 min/day or five or more days of moderate-intensity activity or walking of at least 30 min/day or five or more days of any combination of walking, moderate- or vigorous-intensity activities, that achieved a minimum of at least 600 MET-min/week. Participants who did not meet any of the previous two criteria were classified as having a low level of PA. |
| Sedentary behavior | More than eight hours of sitting per day. |
| Past medical history | Self-reported hypertension, diabetes mellitus, hypercholesterolemia, and cardiovascular disease as diagnosed by a doctor or under current use of anti-hypertensive drugs, anti-diabetics, or lipid-lowering drugs. |
| BMI | BMI was calculated by dividing body weight by the squared height (kg/m$^2$). BMI was categorized based on the WHO guidelines: underweight ($<$ 18.5 kg/m$^2$), normal (18.5–24.9 kg/m$^2$), overweight (25.0–29.9 kg/m$^2$), and obesity ($\geq$ 30 kg/m$^2$). |
| Abdominal obesity | WC $\geq$ 90 cm among men or $\geq$ 80 cm among women |
| Elevated FBG | FBG $\geq$ 5.6 mmol/L |
| Low HDL-C | HDL-C < 1.03 mmol/L in men and < 1.29 mmol/L in women |
| Elevated TG | Serum TG $\geq$ 1.7 mmol/L |
| Hypertension | Systolic BP $\geq$ 130 mmHg or diastolic BP $\geq$ 85 mmHg |

WC, waist circumference; BP, blood pressure; FBG, fasting blood glucose; TG, triglyceride; HDL-C, high-density lipoprotein cholesterol; PA, physical activity; GPAQ, global physical activity questionnaire; MET, metabolic equivalent of task; BMI, body mass index.

value was less than 0.05 and removed when the P-value was over 0.1. A correlation matrix was created to confirm the absence of a strong correlation ($\gamma$ > 0.80) among the independent variables. The result of the Hosmer-Lemeshow test of each model was P > 0.05. P-value < 0.05 was considered statistically significant. Data analyses were performed using SPSS version 28.0 (IBM SPSS Inc, Armonk, NY, USA).

## Ethical considerations

To ensure participant rights, the STEPS survey steering committee developed an ethical protocol, which was approved by the Ethical Committee of the Ministry of Health of Mongolia on June 19, 2018 (approval number #53) [20,22]. Written informed consent was obtained from all participants.

## Results

### Characteristics of participants according to sex

In total, 5,695 participants (2,577 males and 3,118 females) with no missing data on MetS measurements were included in the study. The mean age of the participants was 42.4 years, with 41.6 years for males and 43.0 years for females. Most participants belonged to the Khalkh ethnic group (84.8%), lived in urban areas (64.2%), particularly in Ulaanbaatar City (41.6%), had secondary or higher education (90.2%), were married at the time of data collection (73.1%), and had a monthly income of 500,000 MNT (approximately 144 USD) or more (64.1%) (Table 2). Current smokers and alcohol drinkers accounted for 25.5% and 37.6% of the participants, respectively. Only 29.6% had sufficient intake of fruits and vegetables. More than two-thirds (69.8%) of the participants reported performing high or moderate levels of PA, whereas 7.5% reported having a sedentary lifestyle. In terms of medical history, 32.1% had HT, 5.2% had DM, 6.4% had HCE, and 16.5% had CVD. More than half of the participants (59.1%) were categorized into 'overweight' or 'obesity'. Compared to male participants, female participants had a significantly higher percentage of participants who were 45–69 years, belonged to the Khalkh ethnic group, lived in urban areas or Ulaanbaatar, had education at college or higher, were divorced/widowed/separated, unemployed, and had a monthly income of 300,000- < 1,000,000 MNT. In terms of health behavior and lifestyle, males had more participants who were current smokers, were current alcohol users, and had insufficient fruit and vegetable intake than females. Females had more participants who had a history of HT, HCE, and CVD, whose BMI was categorized into obesity, and who had abdominal obesity than males. Characteristics of all participants according to residence, BMI, and abdominal obesity are shown in S1–S3 Tables.

### Prevalence of MetS and its components

The prevalence of MetS was 37.4% (n = 2,128) in total, 35.2% (n = 906) in men, and 39.2% (n = 1,222) in women (Fig 1). The MetS prevalence was significantly higher in women than in men (P = 0.002). The most prevalent component of MetS was abdominal obesity (62.9%) followed by elevated FBG (53.9%), HT (43.9%), elevated TG (33.2%), and low HDL-C levels (26.8%). The most prevalent component of MetS was elevated FBG level (58.7%) among men and abdominal obesity (74.2%) among women. The prevalence of abdominal obesity and low HDL-C level was significantly higher and that of elevated FBG, elevated TG, and HT was significantly lower in women than in men (P < 0.001).

Of the 5,695 participants, 5.1% (5.5% of men and 4.7% of women) had all five components (Fig 2). The percentage of participants who did not have any component of MetS was higher in men (12.3%) than women (9.3%, P < 0.001) and that of participants who had two components was lower in men (24.5%) than women (27.5%, P = 0.0097).

### Factors associated with MetS among total participants

The prevalence of MetS in each category of all variables is shown in Table 3. All participants who were categorized into MetS met the criteria of abdominal obesity. Multivariate logistic regression analysis with forced-entry was used to identify the factors associated with MetS. Among the total participants, the factors associated with MetS were the age groups of 30–45 years (AOR = 1.47, 95% CI 1.16–1.87) and 45–69 years (AOR = 2.24, 95% CI 1.76–2.86) compared to the 18–29 year group; Buryat ethnicity (AOR = 1.88, 95% CI 1.20–2.95) compared to Khalkh ethnicity; living in the central region (AOR = 1.37, 95% CI 1.01–1.86) and Ulaanbaatar City (AOR = 1.48, 95% CI 1.11–1.99) compared to the western region; a monthly

**Table 2. Characteristics of participants according to sex (N = 5,695).**

| Variables | Total (N = 5695) | Male (N = 2577) | Female (N = 3118) | P-value[c] |
|---|---|---|---|---|
| | N (%) | n (%) | n (%) | |
| **Age group (years)** | | | | 0.002 |
| 18–29 | 1085 (19.1) | 510 (19.8) | 575 (18.4) | |
| 30–44 | 2167 (38.1) | 1027 (39.9) | 1140 (36.6) | |
| 45–69 | 2443 (42.9) | 1040 (40.4) | 1403 (45.0) | |
| **Ethnicity (N = 5669)** | | | | 0.002 |
| Khalkh | 4807 (84.8) | 2126 (82.9) | 2681 (86.4) | |
| Kazakh | 174 (3.1) | 97 (3.8) | 77 (2.5) | |
| Durvud | 240 (4.2) | 111 (4.3) | 129 (4.2) | |
| Buryat | 157 (2.8) | 83 (3.2) | 74 (2.4) | |
| Other | 291 (5.1) | 148 (5.8) | 143 (4.6) | |
| **Residence** | | | | <0.001 |
| Rural | 2039 (35.8) | 1015 (39.4) | 1024 (32.8) | |
| Urban | 3656 (64.2) | 1562 (60.6) | 2094 (67.2) | |
| **Region** | | | | <0.001 |
| Western region | 717 (12.6) | 348 (13.5) | 369 (11.8) | |
| Eastern region | 592 (10.4) | 293 (11.4) | 299 (9.6) | |
| Khangai region | 1094 (19.2) | 535 (20.8) | 559 (17.9) | |
| Central region | 921 (16.2) | 439 (17.0) | 483 (15.5) | |
| Ulaanbaatar | 2371 (41.6) | 962 (37.3) | 1409 (45.2) | |
| **Education (N = 5694)** | | | | <0.001 |
| None | 218 (3.8) | 126 (4.9) | 92 (3.0) | |
| Primary | 344 (6.0) | 195 (7.6) | 149 (4.8) | |
| Secondary | 2567 (45.1) | 1252 (48.6) | 1315 (42.2) | |
| College ≤ | 2565 (45.0) | 1003 (38.9) | 1562 (50.1) | |
| **Marital status (N = 5688)** | | | | <0.001 |
| Never married | 941 (16.5) | 458 (17.8) | 483 (15.5) | |
| Married | 4160 (73.1) | 1936 (75.2) | 2224 (71.4) | |
| Other[a] | 587 (10.3) | 180 (7.0) | 407 (13.1) | |
| **Employment (N = 5575)** | | | | <0.001 |
| Full-time | 2134 (38.3) | 907 (36.2) | 1227 (40.0) | |
| Part-time | 1743 (31.3) | 1029 (41.1) | 714 (23.2) | |
| Unemployed[b] | 1698 (30.5) | 568 (22.7) | 1130 (36.8) | |
| **Monthly income (×1000 MNT)** | | | | <0.001 |
| <100 | 872 (15.3) | 452 (17.5) | 420 (13.5) | |
| 100-<300 | 641 (11.3) | 292 (11.3) | 349 (11.2) | |
| 300-<500 | 527 (9.3) | 212 (8.2) | 315 (10.1) | |
| 500-<1000 | 2235 (39.2) | 948 (36.8) | 1287 (41.3) | |
| 1000≤ | 1420 (24.9) | 673 (26.1) | 747 (24.0) | |
| **Currently smoking** | | | | <0.001 |
| No | 4243 (74.5) | 1318 (51.1) | 2925 (93.8) | |
| Yes | 1452 (25.5) | 1259 (48.9) | 193 (6.2) | |
| **Currently drinking** | | | | <0.001 |
| No | 3551 (62.4) | 1279 (49.6) | 2272 (72.9) | |
| Yes | 2144 (37.6) | 1298 (50.4) | 846 (27.1) | |
| **Insufficient fruit and vegetable intake (N = 5455)** | | | | <0.001 |
| No | 1614 (29.6) | 621 (25.4) | 993 (33.0) | |

*(Continued)*

**Table 2.** (Continued)

| Variables | Total (N = 5695) | Male (N = 2577) | Female (N = 3118) | P-value[c] |
|---|---|---|---|---|
| | N (%) | n (%) | n (%) | |
| Yes | 3841 (70.4) | 1821 (74.6) | 2020 (67.0) | |
| **Level of physical activity (N = 5607)** | | | | <0.001 |
| High | 1450 (25.9) | 812 (32.1) | 638 (20.7) | |
| Moderate | 2464 (43.9) | 975 (38.5) | 1489 (48.4) | |
| Low | 1693 (30.2) | 744 (29.4) | 949 (30.9) | |
| **Sedentary behavior** | | | | 0.975 |
| No | 5270 (92.5) | 2385 (92.5) | 2885 (92.5) | |
| Yes | 425 (7.5) | 192 (7.5) | 233 (7.5) | |
| **History of HT** | | | | <0.001 |
| No | 3865 (67.9) | 1848 (71.7) | 2017 (64.7) | |
| Yes | 1830 (32.1) | 729 (28.3) | 1101 (35.3) | |
| **History of DM** | | | | 0.290 |
| No | 5397 (94.8) | 2451 (95.1) | 2946 (94.5) | |
| Yes | 298 (5.2) | 126 (4.9) | 172 (5.5) | |
| **History of HCE** | | | | <0.001 |
| No | 5333 (93.6) | 2469 (95.8) | 2864 (91.9) | |
| Yes | 362 (6.4) | 108 (4.2) | 254 (8.1) | |
| **History of CVD** | | | | 0.001 |
| No | 4757 (83.5) | 2198 (85.3) | 2559 (82.1) | |
| Yes | 938 (16.5) | 379 (14.7) | 559 (17.9) | |
| **Body mass index (N = 5694)** | | | | <0.001 |
| Normal | 2190 (38.5) | 1070 (41.5) | 1120 (35.9) | |
| Underweight | 137 (2.4) | 71 (2.8) | 66 (2.1) | |
| Overweight | 2041 (35.8) | 927 (36.0) | 1114 (35.7) | |
| Obesity | 1326 (23.3) | 509 (19.8) | 817 (26.2) | |
| **Abdominal obesity** | | | | <0.001 |
| No | 2110 (37.1) | 1305 (50.6) | 805 (25.8) | |
| Yes | 3585 (62.9) | 1272 (49.4) | 2313 (74.2) | |

MNT, Mongolian tugrik; HT, hypertension; DM, diabetes mellitus; HCE, hypercholesterolemia; CVD, cardiovascular disease.

[a]Other includes divorced, widowed, and separated.

[b]Unemployed includes a student, a retired person, and an unemployed person.

[c]A chi-square test was performed.

1 USD = 3,481.66 MNT on April 30, 2023.

income $\geq$ 100,000 MNT compared to a monthly income < 100,000 MNT; moderate (AOR = 1.38, 95% CI 1.15–1.66) or low PA level (AOR = 1.77, 95% CI 1.45–2.15) compared to high PA level; history of HT (AOR = 1.93, 95% CI 1.66–2.24); history of DM (AOR = 2.22, 95% CI 1.61–3.05); and overweight (AOR = 6.53, 95% CI 5.46–7.80) and obesity (AOR = 16.11, 95% CI 13.19–19.67) compared to normal BMI (Table 3). Secondary education compared to no education (AOR = 0.63, 95% CI 0.42–0.96) and insufficient fruit and vegetable consumption (AOR = 0.83, 95% CI 0.71–0.96) were negatively associated with MetS.

## Factors associated with MetS among sex groups

Factors associated with MetS among men were the age groups of 30–45 years (AOR = 1.57, 95% CI 1.06–2.33) and 45–69 years (AOR = 1.96, 95% CI 1.30–2.95) compared to the age

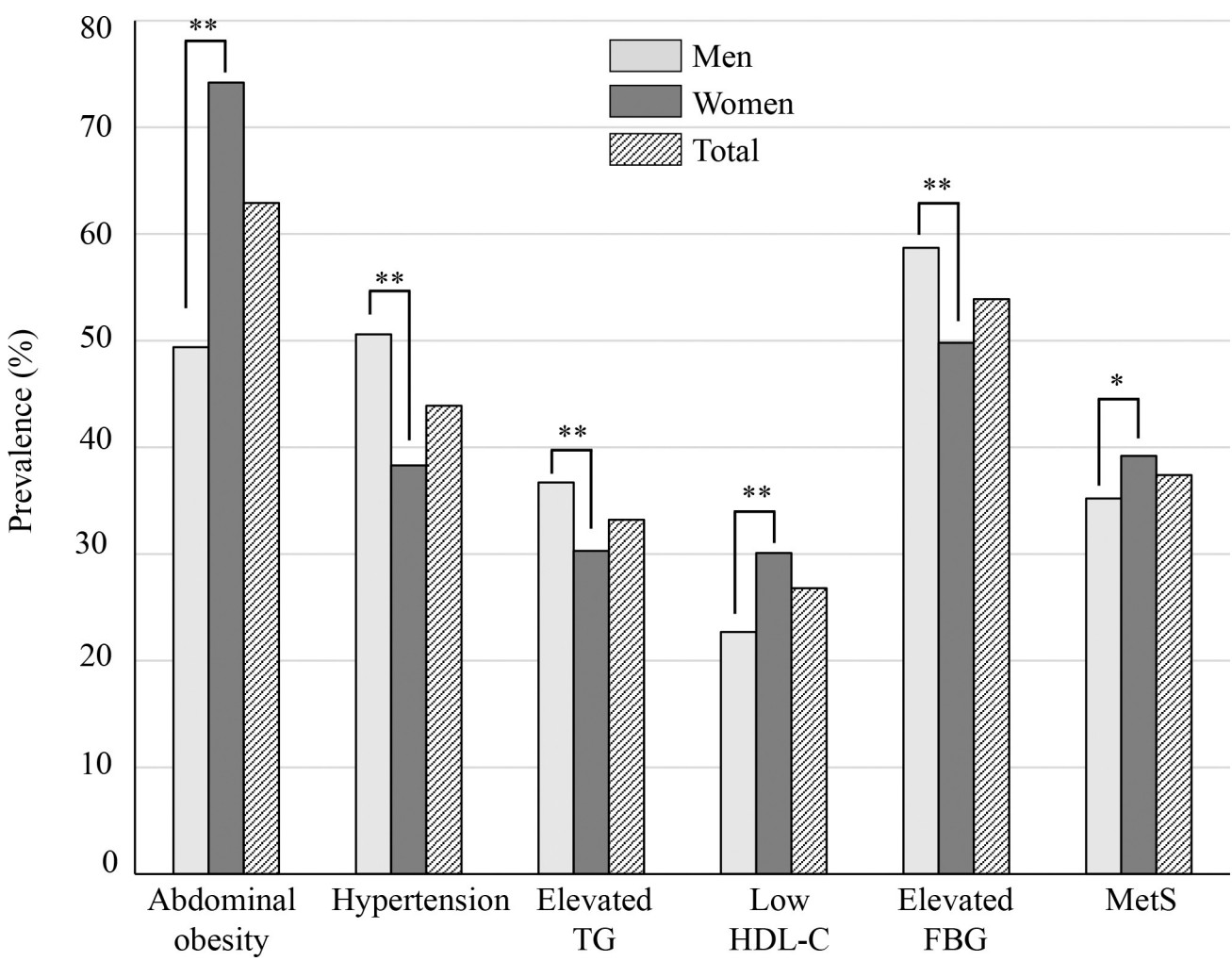

**Fig 1. The prevalence of metabolic syndrome and each component.** The prevalence of metabolic syndrome (MetS) and each component of MetS was compared between men and women by a chi-square test among men and women. TG, triglyceride; HDL-C; high-density lipoprotein cholesterol; FBG, fasting blood glucose; MetS, metabolic syndrome. *P < 0.01; **P < 0.001.

group of 18–29 years; Buryat ethnicity compared to Khalkh ethnicity (AOR = 2.32, 95% CI 1.13–4.80); living in Ulaanbaatar City compared to the western region (AOR = 1.67, 95% CI 1.02–2.72); a monthly income of 100,000– <300,000 MNT (AOR = 1.65, 95% CI 1.03–2.66) or ≥ 1,000,000 MNT (AOR = 1.50, 95% CI 1.00–2.24) compared to a monthly income < 100,000 MNT; currently drinking (AOR = 1.30, 95% CI 1.03–1.65); moderate (AOR = 1.46, 95% CI 1.10–1.95) or low PA level (AOR = 1.73, 95% CI 1.26–2.36) compared to high PA level); history of HT (AOR = 1.85, 95% CI 1.44–2.39), history of DM (AOR = 3.19, 95% CI 1.79–5.66); overweight (AOR = 16.98, 95% CI 12.16–23.72) or obesity (AOR = 56.10, 95% CI 38.33–82.10) compared to normal BMI (Table 3).

Among women, factors associated with MetS were the age groups of 30–45 years (AOR = 1.48, 95% CI 1.09–2.03) and 45–69 years (AOR = 2.65, 95% CI 1.94–3.63) compared to the 18–29 years group; currently smoking (AOR = 1.55, 95% CI 1.08–2.23); moderate (AOR = 1.32, 95% CI 1.04–1.68) or low PA level (AOR = 1.69, 95% CI 1.30–2.20) compared to high PA level; history of HT (AOR = 1.99, 95% CI 1.64–2.42); history of DM (AOR = 1.88, 95% CI 1.27–2.80); overweight (AOR = 3.63, 95% CI 2.90–4.53) or obesity (AOR = 7.92, 95%

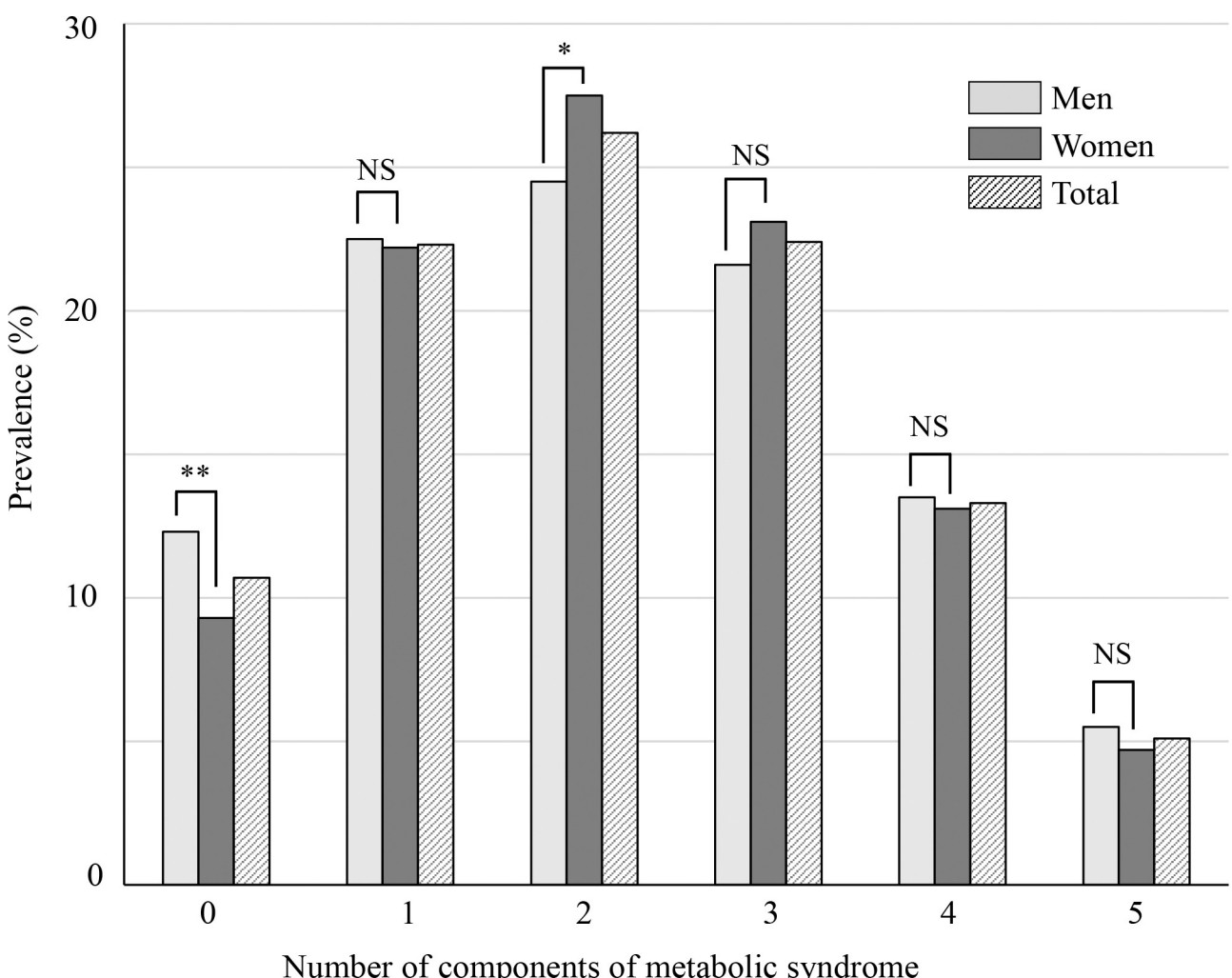

**Fig 2. The number of components of metabolic syndrome.** The prevalence of having components of metabolic syndrome was compared between men and women by a chi-square test. NS, not significant. *P < 0.01; **P < 0.001.

CI 6.21–10.10) compared to normal BMI. Insufficient fruit and vegetable intake was negatively associated with MetS compared to sufficient intake (AOR = 0.82, 95% CI 0.67–0.99) (Table 3).

The results of multivariate logistic regression analyses for MetS among all participants, men, and women using three models are shown in S4–S6 Tables. Older age (≥ 30 years old), low and moderate PA level, history of HT and DM, high BMI (overweight and obesity) were associated with MetS among all participants, men, and women in all three models. In addition, factors associated with MetS in the three models were Buriad ethnicity, residence in the central region and Ulaanbaatar, sufficient fruit and vegetable intake among all participants, current alcohol drinkers among men, and current smokers and sufficient fruit and vegetable intake among women.

## Discussion

In this study, the prevalence of MetS was 37.4% in the general adult population of Mongolia. The MetS prevalence was significantly higher in women than men. The most prevalent MetS component was abdominal obesity in women but elevated FBG in men. Factors associated

**Table 3. The prevalence and factors associated with metabolic syndrome among sex groups (N = 5,695).**

| Variables | Total (N = 5695) | | | Male (N = 2577) | | | Female (N = 3118) | | |
|---|---|---|---|---|---|---|---|---|---|
| | MetS (%) | OR[c] (95% CI) | AOR[d] (95% CI) | MetS (%) | OR[c] (95% CI) | AOR[e] (95% CI) | MetS (%) | OR[c] (95% CI) | AOR[e] (95% CI) |
| **Sex** | | | | | | | | | |
| Male | 35.2 | 1 (Reference) | 1 (Reference) | - | - | - | - | - | - |
| Female | 39.2 | 1.19 (1.07–1.32)** | 0.94 (0.80–1.10) | - | - | - | - | - | - |
| **Age group (years)** | | | | | | | | | |
| 18–29 | 16.8 | 1 (Reference) | 1 (Reference) | 17.3 | 1 (Reference) | 1 (Reference) | 16.5 | 1 (Reference) | 1 (Reference) |
| 30–45 | 33.5 | 2.51 (2.09–3.00)*** | 1.47 (1.16–1.87)** | 35.5 | 2.68 (2.06–3.49)*** | 1.57 (1.06–2.33)* | 31.8 | 2.35 (1.83–3.03)*** | 1.48 (1.09–2.03)* |
| 45–69 | 49.9 | 4.94 (4.14–5.90)*** | 2.24 (1.76–2.86)*** | 43.7 | 3.77 (2.90–4.89)*** | 1.96 (1.30–2.95)** | 54.5 | 6.06 (4.75–7.73)*** | 2.65 (1.94–3.63)*** |
| **Ethnicity** | | | | | | | | | |
| Khalkh | 38.2 | 1 (Reference) | 1 (Reference) | 35.7 | 1 (Reference) | 1 (Reference) | 40.2 | 1 (Reference) | 1 (Reference) |
| Kazak | 28.2 | 0.63 (0.45–0.89)** | 1.47 (0.92–2.33) | 25.8 | 0.63 (0.39–0.99)* | 1.54 (0.75–3.17) | 31.2 | 0.67 (0.41–1.10) | 1.45 (0.77–2.74) |
| Durvud | 28.3 | 0.64 (0.48–0.85)** | 0.75 (0.52–1.09) | 29.7 | 0.76 (0.50–1.16) | 0.86 (0.47–1.57) | 27.1 | 0.55 (0.37–0.82)** | 0.65 (0.40–1.06) |
| Buriad | 48.4 | 1.52 (1.10–2.09)** | 1.88 (1.20–2.95)** | 45.8 | 1.52 (0.98–2.36) | 2.32 (1.13–4.80)* | 51.4 | 1.57 (0.99–2.49) | 1.62 (0.89–2.93) |
| Other | 31.3 | 0.74 (0.57–0.95)* | 1.01 (0.72–1.42) | 32.4 | 0.87 (0.61–1.23) | 1.40 (0.82–2.39) | 30.1 | 0.64 (0.44–0.92)* | 0.78 (0.48–1.25) |
| **Residence** | | | | | | | | | |
| Rural | 34.7 | 1 (Reference) | 1 (Reference) | 31.1 | 1 (Reference) | 1 (Reference) | 38.2 | 1 (Reference) | 1 (Reference) |
| Urban | 38.9 | 1.20 (1.07–1.34)** | 0.83 (0.69–1.01) | 37.8 | 1.34 (1.14–1.59)** | 0.86 (0.63–1.18) | 39.7 | 1.07 (0.91–1.24) | 0.81 (0.63–1.05) |
| **Region** | | | | | | | | | |
| Western | 30.7 | 1 (Reference) | 1 (Reference) | 28.7 | 1 (Reference) | 1 (Reference) | 32.5 | 1 (Reference) | 1 (Reference) |
| Eastern | 30.2 | 0.98 (0.77–1.24) | 0.86 (0.61–1.21) | 29.7 | 1.05 (0.74–1.47) | 0.79 (0.45–1.39) | 30.8 | 0.92 (0.66–1.28) | 0.84 (0.54–1.31) |
| Khangai | 32.9 | 1.11 (0.91–1.36) | 1.05 (0.78–1.40) | 27.7 | 0.95 (0.70–1.28) | 0.77 (0.48–1.25) | 37.9 | 1.27 (0.96–1.67) | 1.18 (0.80–1.74) |
| Central | 41.9 | 1.63 (1.33–2.00)*** | 1.37 (1.01–1.86)* | 39.2 | 1.60 (1.18–2.16)** | 1.30 (0.79–2.15) | 44.4 | 1.66 (1.25–2.20)*** | 1.39 (0.93–2.06) |
| Ulaanbaatar | 41.5 | 1.6 (1.34–1.91)*** | 1.48 (1.11–1.99)** | 41.5 | 1.76 (1.35–2.29)*** | 1.67 (1.02–2.71)* | 41.4 | 1.47 (1.15–1.87)** | 1.31 (0.90–1.91) |
| **Education** | | | | | | | | | |
| None | 32.6 | 1 (Reference) | 1 (Reference) | 29.4 | 1 (Reference) | 1 (Reference) | 37.0 | 1 (Reference) | 1 (Reference) |
| Primary | 33.7 | 1.05 (0.73–1.51) | 0.63 (0.39–1.03) | 26.7 | 0.88 (0.53–1.44) | 0.48 (0.23–1.00)* | 43.0 | 1.28 (0.75–2.19) | 0.85 (0.43–1.71) |
| Secondary | 35.7 | 1.15 (0.86–1.55) | 0.63 (0.42–0.96)* | 32.7 | 1.17 (0.78–1.75) | 0.57 (0.31–1.08) | 38.6 | 1.07 (0.69–1.66) | 0.70 (0.39–1.27) |
| College ≤ | 39.9 | 1.38 (1.03–1.85)* | 0.69 (0.45–1.06) | 40.6 | 1.64 (1.10–2.46)** | 0.53 (0.28–1.03) | 39.5 | 1.11 (0.72–1.72) | 0.84 (0.46–1.54) |
| **Marital status** | | | | | | | | | |
| Never married | 21.6 | 1 (Reference) | 1 (Reference) | 19.4 | 1 (Reference) | 1 (Reference) | 23.6 | 1 (Reference) | 1 (Reference) |

*(Continued)*

**Table 3.** (Continued)

| Variables | Total (N = 5695) | | | Male (N = 2577) | | | Female (N = 3118) | | |
|---|---|---|---|---|---|---|---|---|---|
| | MetS (%) | OR[c] (95% CI) | AOR[d] (95% CI) | MetS (%) | OR[c] (95% CI) | AOR[e] (95% CI) | MetS (%) | OR[c] (95% CI) | AOR[e] (95% CI) |
| Married | 40.6 | 2.49 (2.10–2.94)*** | 1.15 (0.91–1.45) | 39.2 | 2.67 (2.08–3.42)*** | 1.16 (0.78–1.72) | 41.9 | 2.34 (1.86–2.93)*** | 1.10 (0.82–1.49) |
| Other[a] | 39.7 | 2.39 (1.91–3.00)*** | 1.08 (0.80–1.47) | 32.2 | 1.97 (1.34–2.91)** | 1.04 (0.60–1.82) | 43.0 | 2.44 (1.83–3.26)*** | 1.01 (0.70–1.47) |
| **Employment** | | | | | | | | | |
| Full-time | 38.9 | 1 (Reference) | 1 (Reference) | 39.1 | 1 (Reference) | 1 (Reference) | 38.7 | 1 (Reference) | 1 (Reference) |
| Part-time | 32.5 | 0.76 (0.66–0.86)*** | 0.88 (0.74–1.05) | 32.3 | 0.74 (0.61–0.89)** | 0.93 (0.71–1.22) | 32.8 | 0.77 (0.64–0.94)** | 0.81 (0.63–1.03) |
| Unemployed[b] | 40.3 | 1.06 (0.93–1.21) | 1.01 (0.84–1.21) | 34.0 | 0.80 (0.64–0.99)** | 1.14 (0.81–1.61) | 43.5 | 1.22 (1.03–1.43)* | 0.96 (0.77–1.21) |
| **Monthly income (×1000 MNT)** | | | | | | | | | |
| <100 | 28.3 | 1 (Reference) | 1 (Reference) | 24.8 | 1 (Reference) | 1 (Reference) | 32.1 | 1 (Reference) | 1 (Reference) |
| 100–<300 | 36.2 | 1.44 (1.15–1.79)*** | 1.53 (1.15–2.04)** | 30.1 | 1.31 (0.94–1.82) | 1.65 (1.03–2.66)* | 41.3 | 1.48 (1.10–1.99)** | 1.41 (0.97–2.05) |
| 300–<500 | 42.9 | 1.90 (1.51–2.38)*** | 1.41 (1.05–1.90)* | 33.5 | 1.52 (1.07–2.18)* | 1.52 (0.92–2.52) | 49.2 | 2.05 (1.51–2.76)*** | 1.34 (0.91–1.96) |
| 500–<1000 | 38.6 | 1.59 (1.34–1.89)*** | 1.29 (1.02–1.63)* | 38.0 | 1.86 (1.45–2.39)*** | 1.45 (1.00–2.10) | 39.1 | 1.35 (1.07–1.71)* | 1.14 (0.84–1.55) |
| 1000 ≤ | 39.4 | 1.65 (1.37–1.98)*** | 1.36 (1.06–1.76)* | 40.9 | 2.10 (1.61–2.73)*** | 1.50 (1.00–2.24)* | 38.2 | 1.30 (1.01–1.68)* | 1.14 (0.82–1.60) |
| **Currently smoking** | | | | | | | | | |
| No | 37.8 | 1 (Reference) | 1 (Reference) | 36.3 | 1 (Reference) | 1 (Reference) | 38.4 | 1 (Reference) | 1 (Reference) |
| Yes | 36.2 | 0.93 (0.82–1.06) | 1.13 (0.94–1.36) | 33.9 | 0.90 (0.76–1.06) | 1.05 (0.84–1.33) | 50.8 | 1.65 (1.24–2.21)** | 1.55 (1.08–2.23)* |
| **Currently drinking** | | | | | | | | | |
| No | 36.5 | 1 (Reference) | 1 (Reference) | 32.4 | 1 (Reference) | 1 (Reference) | 38.8 | 1 (Reference) | 1 (Reference) |
| Yes | 38.8 | 1.10 (0.99–1.23) | 1.13 (0.98–1.32) | 37.9 | 1.28 (1.08–1.50)** | 1.30 (1.03–1.65)* | 40.2 | 1.06 (0.90–1.25) | 1.04 (0.85–1.28) |
| **Insufficient fruit and vegetable intake** | | | | | | | | | |
| No | 39.0 | 1 (Reference) | 1 (Reference) | 37.0 | 1 (Reference) | 1 (Reference) | 40.3 | 1 (Reference) | 1 (Reference) |
| Yes | 36.7 | 0.91 (0.80–1.02) | 0.83 (0.71–0.96)* | 34.4 | 0.89 (0.74–1.08) | 0.86 (0.66–1.13) | 38.8 | 0.94 (0.80–1.10) | 0.82 (0.67–0.99)* |
| **Physical activity** | | | | | | | | | |
| High | 27.9 | 1 (Reference) | 1 (Reference) | 24.9 | 1 (Reference) | 1 (Reference) | 31.8 | 1 (Reference) | 1 (Reference) |
| Moderate | 37.4 | 1.54 (1.34–1.78)*** | 1.38 (1.15–1.66)*** | 35.5 | 1.66 (1.35–2.04)*** | 1.46 (1.10–1.95)* | 38.7 | 1.35 (1.11–1.65)** | 1.32 (1.04–1.68)* |
| Low | 45.5 | 2.15 (1.85–2.50)*** | 1.77 (1.45–2.15)*** | 46.1 | 2.58 (2.08–3.20)*** | 1.73 (1.26–2.36)*** | 45.0 | 1.75 (1.42–2.16)*** | 1.69 (1.30–2.20)*** |
| **Sedentary lifestyle** | | | | | | | | | |
| No | 37.1 | 1 (Reference) | 1 (Reference) | 34.2 | 1 (Reference) | 1 (Reference) | 39.5 | 1 (Reference) | 1 (Reference) |
| Yes | 40.9 | 1.18 (0.96–1.44) | 1.02 (0.78–1.34) | 47.4 | 1.74 (1.29–2.33)*** | 1.32 (0.85–2.07) | 35.6 | 0.85 (0.64–1.12) | 0.85 (0.59–1.21) |
| **History of HT** | | | | | | | | | |

(*Continued*)

**Table 3.** (Continued)

| Variables | Total (N = 5695) | | | Male (N = 2577) | | | Female (N = 3118) | | |
|---|---|---|---|---|---|---|---|---|---|
| | MetS (%) | OR[c] (95% CI) | AOR[d] (95% CI) | MetS (%) | OR[c] (95% CI) | AOR[e] (95% CI) | MetS (%) | OR[c] (95% CI) | AOR[e] (95% CI) |
| No | 28.5 | 1 (Reference) | 1 (Reference) | 28.2 | 1 (Reference) | 1 (Reference) | 28.7 | 1 (Reference) | 1 (Reference) |
| Yes | 56.2 | 3.22 (2.87–3.62)*** | 1.93 (1.66–2.24)*** | 52.8 | 2.85 (2.39–3.40)*** | 1.85 (1.44–2.39)*** | 58.4 | 3.49 (2.99–4.07)*** | 1.99 (1.64–2.42)*** |
| **History of DM** | | | | | | | | | |
| No | 35.7 | 1 (Reference) | 1 (Reference) | 33.6 | 1 (Reference) | 1 (Reference) | 37.5 | 1 (Reference) | 1 (Reference) |
| Yes | 67.1 | 3.67 (2.87–4.71)*** | 2.22 (1.61–3.05)*** | 65.9 | 3.82 (2.62–5.57)*** | 3.19 (1.79–5.66)*** | 68.0 | 3.54 (2.55–4.93)*** | 1.88 (1.27–2.80)** |
| **History of HCE** | | | | | | | | | |
| No | 35.8 | 1 (Reference) | 1 (Reference) | 34.0 | 1 (Reference) | 1 (Reference) | 37.4 | 1 (Reference) | 1 (Reference) |
| Yes | 60.2 | 2.71 (2.18–3.37)*** | 1.20 (0.91–1.58) | 62.0 | 3.18 (2.13–4.72)*** | 1.16 (0.65–2.05) | 59.4 | 2.45 (1.89–3.19)*** | 1.24 (0.90–1.70) |
| **History of CVD** | | | | | | | | | |
| No | 35.9 | 1 (Reference) | 1 (Reference) | 33.9 | 1 (Reference) | 1 (Reference) | 37.7 | 1 (Reference) | 1 (Reference) |
| Yes | 44.7 | 1.44 (1.25–1.66)*** | 1.13 (0.94–1.37) | 42.5 | 1.44 (1.15–1.80)** | 1.18 (0.84–1.65) | 46.2 | 1.42 (1.18–1.71)*** | 1.12 (0.89–1.41) |
| **Body mass index** | | | | | | | | | |
| Normal | 10.1 | 1 (Reference) | 1 (Reference) | 5.0 | 1 (Reference) | 1 (Reference) | 15.0 | 1 (Reference) | 1 (Reference) |
| Underweight | 5.8 | 0.55 (0.27–1.14) | 0.60 (0.29–1.27) | 4.2 | 0.83 (0.25–2.72) | 0.92 (0.27–3.11) | 7.6 | 0.46 (0.18–1.17) | 0.50 (0.19–1.29) |
| Overweight | 46.7 | 7.77 (6.59–9.15)*** | 6.53 (5.46–7.80)*** | 49.2 | 18.22 (13.46–24.65)*** | 16.98 (12.16–23.72)*** | 44.6 | 4.57 (3.73–5.59)*** | 3.63 (2.90–4.53)*** |
| Obesity | 71.2 | 21.91 (18.25–26.30)*** | 16.11 (13.19–19.67)*** | 77.2 | 63.74 (45.23–89.85)*** | 56.10 (38.33–82.10)*** | 67.4 | 11.41 (9.42–14.62)*** | 7.92 (6.21–10.10)*** |
| **Abdominal obesity** | | | | | | | | | |
| No | 0.0 | - | - | 0.0 | - | - | 0.0 | - | - |
| Yes | 59.4 | - | - | 71.2 | - | - | 52.8 | - | - |

MetS, metabolic syndrome; MNT, Mongolian tugrik; HT, hypertension; DM, diabetes mellitus; HCE, hypercholesterolemia; CVD, cardiovascular disease; OR, odds ratio; AOR, adjusted odds ratio; CI, confidence interval.

[a]Others include divorced, widowed, and separated.

[b]Unemployed includes a student, a retired person, and an unemployed person.

[c]Bivariable logistic regression analysis was performed.

[d]Multivariable logistic regression analysis was performed including all variables in the table other than abdominal obesity (Hosmer-Lemeshow test: P = 0.280).

[e]Multivariable logistic regression analysis was performed including all variables in the table other than sex and abdominal obesity (Hosmer-Lemeshow test: P = 0.607 [men] and P = 0.430 [women]).

*P < 0.05

**P < 0.01

***P < 0.001.

1 USD = 3,481.66 MNT on April 30, 2023.

with MetS among both men and women were older age, low and moderate PA level, history of HT and DM, high BMI (overweight and obesity), and abdominal obesity. Ethnicity, education, income, currently drinking were the associated factors only among men and smokers and sufficient fruit and vegetable intake were only among women. These results showed that there

were differences in the MetS prevalence, MetS component, and the associated factors among sexes.

The MetS prevalence in this study was higher than that in previous studies: 12.0% among 257 people aged 30–60 years in Ulaanbaatar in 2005 and 32.8% in the general population (n = 1,911) aged 40 years or more without ischemic heart disease in 2015 [12,16]. Kim et al reported that the MetS prevalence among adults 20 years old or older who lived in a rural area of Mongolia was 58.0% in 2019 [17]. However, the study consisted of only 143 people in one rural area, and most participants were women (84.6%) and had comorbidities, such as HT and CVD. There has been only one nationwide study on MetS in Mongolia, which reported that the prevalence of MetS was 32.8% in 2009 [12]. However, the study included adults who were 40 years old or older and had no ischemic heart disease. In our study, the prevalence of MetS was 48.3% among participants who were 40 years old or older (n = 3,110). These results suggest that the prevalence of MetS in the general adult population of Mongolia increased from 2005 to 2019.

In this study, the prevalence of MetS was significantly higher in women than in men. This is consistent with the results of a previous study including the general population in Mongolia that showed that the MetS prevalence was 19.4% in men and 40.6% in women [12]. The reason for the higher MetS prevalence among women in this study may be that more female participants had factors associated with a higher prevalence of MetS than male participants, such as being aged 45–69 years, lower level of physical inactivity, a history of HT, DM, HCE, and CVD, and obesity. When considering the prevalence among participants aged 45–69 years between men and women in this study, the prevalence was higher in women (54.5%) than in men (43.7%). For women, estrogen directly regulates glucose and lipid metabolism, but estrogen levels decline as a result of menopause. Estrogen deficiency causes insulin resistance and proatherogenic lipid accumulation and may increase the prevalence of MetS [23,24]. In individuals with obesity, adipose cells release free fatty acids and cytokines, such as tumor necrosis factor-alpha, which inhibit phosphatidylinositide-3-kinase-dependent signal transduction pathways, resulting in decreased glucose absorption in the liver and skeletal muscles. Consequently, insulin is secreted excessively by the pancreatic beta cells. These conditions may cause an increase in blood glucose levels or possibly DM [25,26].

In this study, the prevalence of MetS components differed between men and women. Women exhibited more abdominal obesity and lower levels of HDL-C than men, whereas more men had HT and elevated TG and FBG levels than did women. These results were consistent with those of a previous study conducted in Mongolia [18]. The increased prevalence of abdominal obesity and low HDL-C in women can be explained by hormonal changes due to menopause [27,28]. Obesity and being overweight are significant risk factors for low HDL-C levels [29,30], and people with abdominal obesity have approximately a four-fold higher risk of having lower HDL-C levels than those with normal weight [31]. Mongolian men have unhealthier diets than women, such as consuming larger-sized meals, consuming more fatty meats, finishing meals quickly, eating late, skipping breakfast, and high alcohol consumption. Additionally, Mongolians have a special eating culture in which the breadwinner must consume the initial sample or bowl of each meal and drinks that contain more oils and fats compared with other family members. In Mongolia, the breadwinners are mostly men; therefore, the prevalence of HT and elevated TG and FBG levels in men was higher than that in women.

Factors associated with MetS in men and women were different except for a few overlapping ones. Factors associated with MetS in only men were Buryat ethnicity, living in Ulaanbaatar, higher monthly income, and currently drinking. The Buryat ethnic group is one of the minor ethnicities in Mongolia, who originally lived in southeastern Siberia [32]. Previous studies reported that ethnicity disparity affects the prevalence of MetS [33,34]. Association between

heavy alcohol consumption and a high prevalence of MetS was found among only men in a previous study in Mongolia [12]. It is suggested that associations between alcohol consumption and MetS may be defined comprehensively by sex, race, and ethnicity [35,36]. Therefore, further studies are needed to understand the genetic link and different ethnic characteristics in the development of MetS.

Factors associated with MetS in only women were smoking and sufficient fruit and vegetable intake. The positive relationship between smoking and developing MetS is controversial [37,38], but smoking can increase the risk of developing abdominal obesity, insulin resistance, and low HDL-C levels [39,40]. In this study, smoking was associated with MetS in women rather than men, because women had a higher prevalence of abdominal obesity and lower HDL-C levels than men. Fruits and vegetables provide protection against MetS [9]. Mongolians most commonly consume potatoes as vegetables, but potatoes are not considered vegetables as they contain a large amount of carbohydrates that the body digests rapidly, causing blood glucose and insulin to surge and then dip [41]. Diets high in potatoes and carbohydrates can contribute to obesity, diabetes, and heart disease over the long term [42,43]. Therefore, an updated classification of fruits and vegetables is needed to exempt potatoes from the vegetable category and to improve public awareness of the nutritional content of fruits and vegetables and the effectiveness of healthy diets to combat MetS.

This study has some limitations. First, this is a cross-sectional study; therefore, the results cannot demonstrate the causal relationship between MetS and the associated factors. However, this study included a large population-based sample and a standardized study assessment, and a sample selection technique was used to minimize bias. Second, unmeasured or unselected confounding factors, such as epigenetic factors, and environmental factors, including climate change and air pollution, were not accounted for in this study. Dietary assessment was limited in this study, although diet may also play an important role in the development of MetS. Therefore, further longitudinal studies are required to confirm the findings of this study.

## Conclusion

In conclusion, the prevalence of MetS was 37.4% in the Mongolian adult population and was higher in women than in men. Significant differences were observed between men and women in terms of the prevalence of MetS components and factors associated with MetS. To reduce the high burden of MetS in the population, establishing sex-specific MetS prevention programs that focus on health behaviors, such as reducing alcohol consumption in men and smoking in women, is essential.

## Supporting information

**S1 Table. Characteristics of participants according to residence (N = 5,695).**
(DOCX)

**S2 Table. Characteristics of participants according to body mass index (N = 5,694).**
(DOCX)

**S3 Table. Characteristics of participants according to abdominal obesity (N = 5,695).**
(DOCX)

**S4 Table. Factors associated with metabolic syndrome among all participants (N = 5,695).**
(DOCX)

**S5 Table. Factors associated with metabolic syndrome among men (N = 2,577).**
(DOCX)

**S6 Table. Factors associated with metabolic syndrome among women (N = 3,188).** (DOCX)

## Acknowledgments

This paper uses data from the Mongolia 2019 STEPS survey, implemented by the Ministry of Health and Public Health Institute with the support of the World Health Organization.

## Author Contributions

**Conceptualization:** Lkham-Erdene Byambadoo, Narantuya Davaakhuu, Eiko Yamamoto.

**Data curation:** Lkham-Erdene Byambadoo, Narantuya Davaakhuu, Unursaikhan Surenjav, Bolormaa Norov, Enkhjargal Tserennadmid.

**Formal analysis:** Lkham-Erdene Byambadoo.

**Methodology:** Lkham-Erdene Byambadoo, Souphalak Inthaphatha, Kimihiro Nishino, Eiko Yamamoto.

**Supervision:** Eiko Yamamoto.

**Writing – original draft:** Lkham-Erdene Byambadoo, Eiko Yamamoto.

**Writing – review & editing:** Lkham-Erdene Byambadoo, Narantuya Davaakhuu, Unursaikhan Surenjav, Bolormaa Norov, Enkhjargal Tserennadmid, Souphalak Inthaphatha, Kimihiro Nishino, Eiko Yamamoto.

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
