## [Decision Letter · Decision Letter 0]

1 Feb 2024

PONE-D-23-37323Sex differences in the prevalence and factors associated with metabolic syndrome in the general population of Mongolia: A cross-sectional studyPLOS ONE

Dear Dr. Yamamoto,

Thank you for submitting your manuscript to PLOS ONE. After careful consideration, we feel that it has merit but does not fully meet PLOS ONE’s publication criteria as it currently stands. Therefore, we invite you to submit a revised version of the manuscript that addresses the points raised during the review process.

We look forward to receiving your revised manuscript.

Kind regards,

Jianhong Zhou

Staff Editor

PLOS ONE

Reviewers' comments:

Reviewer's Responses to Questions

**Comments to the Author**

1. Is the manuscript technically sound, and do the data support the conclusions?

Reviewer #1: Yes

Reviewer #2: Yes

2. Has the statistical analysis been performed appropriately and rigorously? 

Reviewer #1: Yes

Reviewer #2: Yes

3. Have the authors made all data underlying the findings in their manuscript fully available?

Reviewer #1: Yes

Reviewer #2: Yes

4. Is the manuscript presented in an intelligible fashion and written in standard English?

Reviewer #1: Yes

Reviewer #2: Yes

5. Review Comments to the Author

Reviewer #1: Dear Authors,

Although the manuscript is well-written, there are some suggestions to improve its quality.

-Title; It's better to mention STEPS or "a nation-wide study" instead of "a cross-sectional study"

-Abstract, and methods of main manuscript: I suggest exclude data of individuals <18 years old. They categorized as adolescents. So, interpretation of the findings in them can be different from other age groups. If you prefer to remain their data, I think it's better to analysis their data separately from other age-groups.

-Introduction; please update Ref.7.

-Insert date of the study in line 69.

-Method; please insert Ethical Committee approval code.

-Results; It is missed expansion of some abbreviated words in the first use, such as HEC.

-Please insert data of "central obesity" in all Tables.

-For all tables, insert Statistical analysis test in the legend.

-Please merge tables 3, and 4.

-Insert additional Tables, based on "urban/rural", "BMI", and "central obesity", separately.

-Discussion; The first paragraph should address the main findings. So, description of results and comparison with other studies should start from the second paragraph.

-Ref.; nearly half of the Ref. are published on more than 10 years ago. Please update.

-Consider language editing for whole manuscript. In some sentences is observed numerous grammatical errors.

Best Regards,

Reviewer #2: The authors have studied Sex differences in the prevalence and factors associated with metabolic syndrome in the general population of Mongolia in a cross-sectional study. It is a well-designed research in a large sample size that included a wide range of demographic and lifestyle factors in their analysis, which allowed for a more nuanced understanding of the factors that contribute to the development of metabolic syndrome in both men and women.

Although similar research has been done and published, with its increasing prevalence, any additional information that sheds light on this condition is valuable.

There are a few comments that need to be considered:

Criteria used for MetS are described based on (IDF) and Adult Treatment Panel (ATP) for estimating the prevalence of MetS, but in other parts only IDF criteria are used. I think ATP criteria may not be necessary. The paper just can only use IDF critera.

Table 2 provides no more information compared to Table 3. It could be potentially removed.

Table 3, is the adjusted p-value calculated after adjustment with all variables ?(more than 15 variables are introduced)

6. PLOS authors have the option to publish the peer review history of their article (what does this mean?). If published, this will include your full peer review and any attached files.

Reviewer #1: No

Reviewer #2: No

---

## [Author Response · Author response to Decision Letter 0]

1 Mar 2024

We would like to thank the editor and reviewers for reviewing our manuscript. We have revised the manuscript according to your comments, which were very helpful. The revised manuscript has been proofread by a native English speaker. The revisions have been completed and our responses are as follows.

Journal Requirements: 

We have carefully revised our manuscript to meet PLOS ONE’s style requirements including the file names.

Thank you for your suggestion. The data that support the findings of this study are available at the WHO NCD Microdata Repository (https://extranet.who.int/ncdsmicrodata/index.php/catalog).

We have revised the references according to the comment No. 12 by Reviewer 1. The below table shows how we have revised references and reference numbers. We used EndNote (a software for reference manager) and revision of references and reference numbers cannot be retracted using the function of Track Change. Therefore, we have used red color for revised reference number and revised references in the reference section. 

We are very sorry that the reference style did not follow the journal style. We have checked the reference style on the journal site and revised the reference style in the manuscript.

Ref. Original→　Revision

1 Deepa M, 2007 →The IDF consensus worldwide definition, 2006

5 Ford ES, 2005 →Yakoob MY, 2016

7 The IDF consensus worldwide definition, 2006 →Ref. number changed to 1.

Ref. 8-21 →Ref. 7-20

22 NCEP expert panel on detection, evaluation,… , 2002 →Ref. number changed to 26

23 National Center for Public Health, 2020 →Ref. number changed to 21

24 Schneider JT, 2006 →(Ref. 22) Iwasa T, 2023.

25 Regitz-Zagrosek C, 2007 →(Ref. 23) Meloni A, 2023

26 Cornier MA, 2008 →(Ref. 24) Kautzky-Willer A, 2023

27 Kuk JL, 2010 →(Ref. 25) Tramunt B, 2020

28 Morales DD, 2008 →Changed to appropriate reference (Ref. 18)

29 Zago Z, 2004 →(Ref. 27) Li H, 2021

Ref. 30-36　→ Ref. 28-34

37 Fujita N, 2011 →(Ref. 35) Hirakawa M, 2015

38 Alkerwi A, 2009 →Erased due to Ref. 34 and Ref. 35 are enough for the sentence.

39 Yoon YS, 2004 →Erased due to Ref. 34 and Ref. 35 are enough for the sentence.

Ref. 40-43→ Ref. 36-39

44 Halton TL, 2006 →(Ref. 40) Muraki I, 2016

45 Mozaffarian D, 2011 →(Ref. 41) Hojati A, 2024

46 Beulens JW, 2007 →(Ref. 42) Fung TT, 2016

Reviewer #1:

1. Title; It's better to mention STEPS or "a nation-wide study" instead of "a cross-sectional study."

According to the reviewer’s comment and advice of the English proofreader, we have revised the title from “Sex differences in the prevalence and factors associated with metabolic syndrome in the general population of Mongolia: A cross-sectional study” to “Sex differences in the prevalence of metabolic syndrome and associated factors in the general population of Mongolia: A nationwide study.”

2. Abstract, and methods of main manuscript: I suggest exclude data of individuals <18 years old. They categorized as adolescents. So, interpretation of the findings in them can be different from other age groups. If you prefer to remain their data, I think it's better to analysis their data separately from other age-groups.

We have excluded participants who were younger than 18 years old and the revised manuscript includes 5,695 participants who were 18 years old or older.

3. Introduction; please update Ref.7.

Line 50: The reviewer suggested to update the reference of the definition and criteria of metabolic syndrome, but we think that the reference in our manuscript is the latest version (International Diabetes Federation. The IDF consensus worldwide definition of the metabolic syndrome. Brussels, Belgium: International Diabetes Federation; 2006). 

The reference number in Line 50 has been changed from 7 to 1, because we have changed the reference for the first sentence of the introduction section from “Deepa et al, Diabetes Metab Res Rev. 2007” to “The IDF consensus worldwide definition of the metabolic syndrome, 2006.” The first sentence explains about component of metabolic syndrome; therefore, “The IDF consensus worldwide definition of the metabolic syndrome, 2006” is more appropriate than the previous Ref. 1 (Deepa et al, Diabetes Metab Res Rev. 2007).

4. Insert date of the study in line 69

Line 69: We have added the year (2019) of the study as follows, “58.0% in adults ≥20 years old in a rural area in 2019.” 

5. Method; please insert Ethical Committee approval code.

Line 148: We have added the ethical approval number #53 in the sentence. 

6. Results; It is missed expansion of some abbreviated words in the first use, such as HEC.

Line 175: HEC was our mistake and it has been revised to “HCE. We have checked all abbreviations and added the full spelling for an abbreviation when it is used for the first time.

7. Please insert data of "central obesity" in all Tables.

We have added “abdominal obesity (=central obesity)” in Tables 2 and 3. 

8. For all tables, insert Statistical analysis test in the legend.

We have added statistical analysis test in the footnote of Tables 2 and 3.

9. Please merge tables 3, and 4.

We have merged Tables 3 and 4 into one table (Table 3).

10. Insert additional Tables, based on "urban/rural", "BMI", and "central obesity", separately.

We have made three tables based on residence (rural and urban), BMI (normal, underweight, overweight, and obesity), and abdominal obesity (yes and no) and put them as S1-3 Tables in supporting information file. Supporting information captions has been added at the end of the manuscript file.

11. Discussion; The first paragraph should address the main findings. So, description of results and comparison with other studies should start from the second paragraph.

Lines 270-277: We have revised the first paragraph of the discussion section and addressed the main findings as follows, “In this study, the prevalence of MetS was 37.4% in the general adult population of Mongolia. The MetS prevalence was significantly higher in women than men. The most prevalent MetS component was abdominal obesity in women but elevated FBG in men. Factors associated with MetS among both men and women were older age, low and moderate PA level, history of HT and DM, high BMI (overweight and obesity), and abdominal obesity. Ethnicity, education, income, current alcohol consumption were the associated factors only among men and current smoking and sufficient fruit and vegetable intake were only among women. These results showed that there were differences in the MetS prevalence, MetS component, and the associated factors among sexes.”

12. Ref.; nearly half of the Ref. are published on more than 10 years ago. Please update.

We have revised 10 references that were published more than 10 years ago to more recent papers (published in 2015 or after). According to revision of the manuscript, reference numbers have been revised. The below table shows how we have revised references. We used EndNote (a software for reference manager) and revision of references and reference numbers cannot be retracted using the function of Track Change. Therefore, we have used red color for revised reference number and revised references in the reference section.

Ref. Original　→　Revision

1 Deepa M, 2007 →The IDF consensus worldwide definition, 2006

5 Ford ES, 2005 →Yakoob MY, 2016

7 The IDF consensus worldwide definition, 2006 →Ref. number changed to 1.

Ref. 8-21 →Ref. 7-20

22 NCEP expert panel on detection, evaluation,… , 2002 →Ref. number changed to 26

23 National Center for Public Health, 2020 →Ref. number changed to 21

24 Schneider JT, 2006 →(Ref. 22) Iwasa T, 2023.

25 Regitz-Zagrosek C, 2007 →(Ref. 23) Meloni A, 2023

26 Cornier MA, 2008 →(Ref. 24) Kautzky-Willer A, 2023

27 Kuk JL, 2010 →(Ref. 25) Tramunt B, 2020

28 Morales DD, 2008 →Changed to appropriate reference (Ref. 18)

29 Zago Z, 2004 →(Ref. 27) Li H, 2021

Ref. 30-36　→ Ref. 28-34

37 Fujita N, 2011 →(Ref. 35) Hirakawa M, 2015

38 Alkerwi A, 2009 →Erased due to Ref. 34 and Ref. 35 are enough for the sentence.

39 Yoon YS, 2004 →Erased due to Ref. 34 and Ref. 35 are enough for the sentence.

Ref. 40-43→ Ref. 36-39

44 Halton TL, 2006 →(Ref. 40) Muraki I, 2016

45 Mozaffarian D, 2011 →(Ref. 41) Hojati A, 2024

46 Beulens JW, 2007 →(Ref. 42) Fung TT, 2016

13. Consider language editing for whole manuscript. In some sentences is observed numerous grammatical errors.

We have asked a native English speaker who has a PhD to proofread the manuscript and we have revised our manuscript accordingly.

Reviewer #2:

14. Criteria used for MetS are described based on (IDF) and Adult Treatment Panel (ATP) for estimating the prevalence of MetS, but in other parts only IDF criteria are used. I think ATP criteria may not be necessary. The paper just can only use IDF critera.

We have erased the explanation about the criteria of ATP III and results based on the ATP III criteria in our manuscript and figures.

15. Table 2 provides no more information compared to Table 3. It could be potentially removed.

This study focuses on differences between men and women in terms of metabolic syndrome and Table 3 shows the differences in factors associated with metabolic syndrome between the sexes. Table 2 shows the differences in the characteristics between male and female participants, and the characteristics differences were used to explain why differences were found in the prevalence of metabolic syndrome and associated factors between the sexes in the discussion section. Therefore, we would like to keep Table 2 in our manuscript.

16. Table 3, is the adjusted p-value calculated after adjustment with all variables? (more than 15 variables are introduced)

Lines 139-142: We included all variables in Table 3 in multivariable logistic regression analyses. Before performing multivariable analysis, we confirmed there was no absence of a strong correlation among the independent variables. We also confirmed that the result of Hosmer-Lemeshow test was P > 0.05 in each model. We have added the following sentences in the method section, “Multivariable logistic regression analysis was performed on MetS using all the variables other than abdominal obesity. A correlation matrix was created to confirm the absence of strong correlation (γ > 0.80) among the independent variables. The result of the Hosmer-Lemeshow test of each model was P > 0.05. “ 

Lines 239-243: We have also added the result of Hosmer-Lemeshow test in the footnote of Table 3 as follows, “dMultivariable logistic regression analysis was performed including all variables in the table other than abdominal obesity (Hosmer-Lemeshow test: P = 0.280).

eMultivariable logistic regression analysis was performed including all variables in the table other than sex and abdominal obesity (Hosmer-Lemeshow test: P = 0.607 [men] and P = 0.430 [women]).”

---

## [Decision Letter · Decision Letter 1]

11 Jul 2024

PONE-D-23-37323R1Sex differences in the prevalence of metabolic syndrome and associated factors in the general population of Mongolia: A nationwide studyPLOS ONE

Dear Dr. Yamamoto,

Thank you for submitting your manuscript to PLOS ONE. After careful consideration, we feel that it has merit but need substantial revision in terms of English Editing. Therefore, we invite you to submit a revised version of the manuscript that addresses the issue. 

We look forward to receiving your revised manuscript.

Kind regards,

Mithun Sikdar

Academic Editor

PLOS ONE

Journal Requirements:

Additional Editor Comments:

I am ready to accept the paper, provided it is revised with English Editing.

Reviewers' comments:

Reviewer's Responses to Questions

**Comments to the Author**

1. If the authors have adequately addressed your comments raised in a previous round of review and you feel that this manuscript is now acceptable for publication, you may indicate that here to bypass the “Comments to the Author” section, enter your conflict of interest statement in the “Confidential to Editor” section, and submit your "Accept" recommendation.

Reviewer #1: All comments have been addressed

Reviewer #3: All comments have been addressed

2. Is the manuscript technically sound, and do the data support the conclusions?

Reviewer #1: Yes

Reviewer #3: Yes

3. Has the statistical analysis been performed appropriately and rigorously? 

Reviewer #1: Yes

Reviewer #3: Yes

4. Have the authors made all data underlying the findings in their manuscript fully available?

Reviewer #1: Yes

Reviewer #3: Yes

5. Is the manuscript presented in an intelligible fashion and written in standard English?

Reviewer #1: Yes

Reviewer #3: (No Response)

6. Review Comments to the Author

Reviewer #1: Dear Authors,

All comments are adequately addressed, just please re-check sum of new estimations for each column. For example, sum of HTN data is incorrect.

Best Regards,

Reviewer #3: This is a simple straight forward Population Health data with adequate sample size representing the studied Population of Mongolia. Geographically and Ethnically the reported data seems to have the nationally standardized parameters for comparative reference which is in congruence to International classifications when computing for the Metabolic Syndrome.

The overall results suggests no different but general observation and findings that females are more predisposed to MetS as compared to the male counterparts with Abdominal Obesity being more prevalent. Dietary assessment was perhaps the only limitation from this study where diet could have also played a vital role in the analysis of the reported study. Correlation tests and Regression models could have been tried and tested for a more analytical statistical interpretation.

However, in simple text and language it is a good publication.

7. PLOS authors have the option to publish the peer review history of their article (what does this mean?). If published, this will include your full peer review and any attached files.

Reviewer #1: No

Reviewer #3: No

---

## [Author Response · Author response to Decision Letter 1]

21 Aug 2024

Response to the editor’s and reviewers’ comments 

We would like to thank the editor and reviewers for reviewing our manuscript. We have revised the manuscript according to your comments, which were very helpful. The revised manuscript has been proofread by a native English speaker. The revisions have been completed and our responses are as follows.

Journal Requirements: 

We have added a new reference (Ref 21) according to the comment by Reviewer 1. In the manuscript, reference numbers of 21-42 have been revised to 22-43. We used EndNote (a software for reference manager) and revision of references and reference numbers cannot be retracted using the function of Track Change. Therefore, we have used red color for revised reference numbers and revised references in the reference section. 

Additional Editor Comments:

2. I am ready to accept the paper, provided it is revised with English Editing.

The revised manuscript has been English proofread by a native English speaker. Therefore, there are many parts that have been revised according to the advice of the English proofreader.

Reviewer #1:

3. All comments are adequately addressed, just please re-check sum of new estimations for each column. For example, sum of HTN data is incorrect.

Lines 125-130, Table 2, and S1-3 Tables: Our study used the data from 5,695 participants who had no missing values for variables related to the MetS criteria, but some missing values for other variables. We have added the explanation about the missing values in the methods section as follows, “The total number of STEPS participants was 6,654 but 959 participants were excluded due to missing values on MetS measurement data. Finally, the data of 5,695 participants were analyzed in this study. There were missing values about ethnicity (n = 26), education (n = 1), marital status (n = 7), employment (n = 120), fruit and vegetable intake (n = 240), physical activity (n = 88), and body mass index (n = 1). The missing values were treated by multiple imputation method [21].” We have also added the number of participants for variables that had missing data in Table 2 and S1-3 Tables.

Line 112: We have revised the sentence as follow, “….5,695 adult participants of the STEPS who were 18‒69 years old and whose data had no missing values for variables related to the MetS criteria.”

We are very sorry that there were mistakes of numbers and percentages in tables. We have carefully checked all numbers and percentages in all tables and revised them correctly. 

Reviewer #3:

4. Dietary assessment was perhaps the only limitation from this study where diet could have also played a vital role in the analysis of the reported study. 

Lines 361-362: We have added a sentence in the paragraph of limitations as follow, “Dietary assessment was limited in this study, although diet may also play an important role in the development of MetS.”

5. Correlation tests and Regression models could have been tried and tested for a more analytical statistical interpretation. 

S4-6 Tables: According to the comment, we have performed two additional models of multivariate logistic regression, namely stepwise forward-selection and backward-selection. We have added three tables to show factors associated with MetS among all participants (S4 Table), men (S5 Table), and women (S6 Table) using the three models. 

Lines 143-147: In the methods section, we have added as follows, “In multivariable logistic regression analysis on MetS, three models were performed. In Model 1 (the forced-entry method), all the variables other than abdominal obesity were included. In Models 2 and 3 (the forward and backward stepwise selection method), variables were added when the P-value was less than 0.05 and removed when the P-value was over 0.1.“ 

Lines 272-279: In the results section, we have added as follows, “The results of multivariate logistic regression analyses for MetS among all participants, men, and women using three models are shown in S4-6 Tables. Older age (≥ 30 years old), low and moderate PA level, history of HT and DM, high BMI (overweight and obesity) were associated with MetS among all participants, men, and women in all three models. In addition, factors associated with MetS in the three models were Buriad ethnicity, residence in the central region and Ulaanbaatar, sufficient fruit and vegetable intake among all participants, current alcohol drinkers among men, and current smokers and sufficient fruit and vegetable intake among women.”

---

## [Editor Report · Decision Letter 2]

18 Sep 2024

Sex differences in the prevalence of metabolic syndrome and associated factors in the general population of Mongolia: A nationwide study

PONE-D-23-37323R2

Dear Dr. Yamamoto

We’re pleased to inform you that your manuscript has been judged scientifically suitable for publication and will be formally accepted for publication once it meets all outstanding technical requirements.

Kind regards,

Mithun Sikdar

Academic Editor

PLOS ONE

---

## [Editor Report · Acceptance letter]

11 Oct 2024

PONE-D-23-37323R2 

PLOS ONE

Dear Dr. Yamamoto, 

I'm pleased to inform you that your manuscript has been deemed suitable for publication in PLOS ONE. Congratulations! Your manuscript is now being handed over to our production team.

Kind regards, 

on behalf of

Dr. Mithun Sikdar 

Academic Editor

PLOS ONE